# Micrometer Copper-Zinc Alloy Particles-Reinforced Wood Plastic Composites with High Gloss and Antibacterial Properties for 3D Printing

**DOI:** 10.3390/polym12030621

**Published:** 2020-03-09

**Authors:** Feiwen Yang, Jianhui Zeng, Haibo Long, Jialin Xiao, Ying Luo, Jin Gu, Wuyi Zhou, Yen Wei, Xianming Dong

**Affiliations:** 1Biomass 3D Printing Materials Research Center, College of Materials and Energy, South China Agricultural University, Guangzhou 510642, China; yangfeiwen163@163.com (F.Y.); i992595479@163.com (J.Z.); longhaibo123456@163.com (H.L.); q625757497@163.com (J.X.); luoying@scau.edu.cn (Y.L.); gujin57@163.com (J.G.); 2Guangdong Laboratory of Lingnan Modern Agricultural Science and Technology, South China Agricultural University, Guangzhou 510642, China; 3Key Laboratory for Modern Agriculture Materials of Ministry of Education, South China Agricultural University, Guangzhou 510642, China; 4Department of Chemistry and the Tsinghua Center for Frotier Polymer Research, Tsinghua University, Beijing 10084, China; weiyen@mail.tsinghua.edu.cn

**Keywords:** copper-zinc alloy, wood plastic composites, mechanical properties, antibacterial property, 3D printing

## Abstract

In this work, micrometer copper-zinc alloy particles-reinforced particleboard wood flour/poly (lactic acid) (mCu-Zn/PWF/PLA) wood plastic composites with high gloss and antibacterial properties for 3D printing were prepared by a melt blending process. The structure and properties of the composites with different contents of mCu-Zn were analyzed by means of mechanical testing, dynamic mechanical analysis, thermogravimetric analysis, differential scanning calorimetry, X-ray diffraction, scanning electron microscopy, and antibacterial testing. The results showed that the mechanical properties, thermal stability, and antibacterial performance of the composites were significantly improved, as mCu-Zn was added into the wood plastic composites. When adding 2 wt.% mCu-Zn, the flexural strength of mCu-Zn/PWF/PLA composites (with 5 wt.% of particleboard wood flour) (PWF) increased by 47.1% compared with pure poly (lactic acid) (PLA), and 18.9% compared with PWF/PLA wood plastic composites. The surface gloss was increased by 1142.6% compared with PWF/PLA wood plastic composites. Furthermore, the inhibition rates of mCu-Zn/PWF/PLA composites against Escherichia coli reached 90.43%. Therefore, this novel high gloss and antibacterial wood plastic composites for fused deposition modeling (FDM) 3D printing have potential applications in personalized and classic furniture, art, toys, etc.

## 1. Introduction

Wood plastic composites have been widely known for decades and are widely used in furniture, park benches, fences, and door and window frames due to the advantages of low cost, corrosion resistance, insect resistance, and long life compared with natural wood [1,2,3]. Poly (lactic acid) is a biodegradable polymer with similar properties to many petroleum-based plastics. Therefore, the preparation of poly (lactic acid)-based wood plastic composites is an attractive research topic, since the composites have a similar appearance to wood, excellent mechanical properties, and are fully degradable [4]. Balar et al. developed bio-based thermoplastic composites of poly (lactic acid) and hazelnut shell waste and evaluated the effect of hazelnut content on the thermal and thermomechanical properties of wood plastic composites. The composites are fully degradable, dimensionally stable and have excellent mechanical properties, and can be used as fences, trims, flooring, automotive interior parts, and furniture [5]. However, many wood plastic composites have poor thermal properties, rough and dull appearance (surface coating is required), and poor antibacterial effects.

After the 21st century, the demand for particleboard has increased year by year, and the particleboard industry has entered a stage of innovation and stable development [6]. However, most particleboards contain formaldehyde, which may be released after installation [7]. In addition, the boards cannot be completely degraded in a short period of time. The smoke generated during the combustion of particleboard is also mostly formaldehyde, which has a negative impact on the environment [8]. Therefore, it is very meaningful to efficiently recycle particleboard for the production of wood plastic composites. It has been reported that particleboard waste fibers are combined with high density polyethylene to prepare recyclable and durable products [9].

With the increasing awareness of health and hygiene in modern society, antibacterial polymer composites have become one of the most studied materials in the past decade [10]. Ag^+^ and silver nanoparticles are the most commonly used metal antibacterial agents for the preparation of antibacterial polymer composites because of their high antibacterial activity, chemical stability, and low toxicity. Fortunati et al. successfully prepared binary and ternary poly (lactic acid) (PLA) nanocomposite films containing cellulose nanocrystals and silver nanoparticles by a twin screw extrusion process. They proved that the PLA nanocomposite films have a bactericidal effect on Staphylococcus aureus (*S. aureus*) and Escherichia coli (*E. coli*) [11]. In fact, compared with silver, copper-zinc (Cu-Zn) binary alloys offer their own unique advantages, including bright colors, high corrosion resistance, low cost, and good antibacterial activity. Tripathi et al. reported bio-composite scaffolds containing chitosan, nano-hydroxyapatite, and Cu-Zn alloy nanoparticles by the freeze-drying technique [12]. Antibacterial biopolymer fibers containing copper and zinc nanoparticles prepared by electrospinning were also reported [13]. 

Additive manufacturing, commonly known as 3D printing, builds objects by layer-by-layer printing based on digital model files [14]. In recent years, 3D printing technology has begun to evolve from rapid prototyping technology to rapid manufacturing methods, and is gradually being used in industrial production [15]. Fused deposition modeling (FDM) is one of the most commonly used 3D printing technologies [16]. Preparation of coconut fiber/poly (lactic acid) (PLA) bio-composites by FDM 3D printing technology has been studied. Incorporation of coconut fiber and changing the microstructure of the bio-composites through printing width has been introduced to improve its mechanical properties [17]. Pringle et al. researched the mixing of PLA pellets with recycled wood waste to produce wood plastic composites for the furniture industry through 3D printing [6]. In fact, Mazzanti et al. comprehensively analyzed the effects of additive formulations and processing parameters on the mechanical properties of biofilled 3D printed samples, including printing speed, extruder and bed temperatures, nozzle diameter, and polymer matrix types. Their results show that the mechanical properties of acrylonitrile-butadiene-styrene (ABS) and poly (lactic acid) (PLA) cannot benefit from biofillers, while polyolefins have become more performant. Moreover, it is likely that better quality biocomposite printing might be obtained with specifically designed printing machines [18].

In order to develop antibacterial wood plastic composites as FDM 3D printing materials, improve the surface finish of 3D printed works, and reduce the environmental pollution of waste particleboard, in the present study, novel wood plastic composites were prepared by using poly (lactic acid) as a matrix, and particleboard wood flour (PWF) and micrometer copper-zinc alloy particles (mCu-Zn) as reinforcements by a melt blending process. The purpose is to obtain mCu-Zn-reinforced wood plastic composites with good mechanical, thermal, antibacterial, and printing performance as FDM 3D printing materials. Figure 1 shows the flowchart of preparation and characterization of micrometer copper-zinc alloy particles-reinforced particleboard wood flour/poly (lactic acid) (mCu-Zn/PWF/PLA) composites for 3D printing. 

## 2. Experiment

### 2.1. Materials

Poly (lactic acid) (PLA, 4032D), with a density of 1.24 g/cm^3^ and melt flow index (MFI) of 7 g/10 min (at 210 °C, load 2.16 kg), was supplied by NatureWorks (Blair, Nebraska, NE, USA). Particleboard wood flour (PWF) was from furniture waste, provided by Guangzhou Sophia Home Furnishing Co., Ltd (Guangzhou, China). Micrometer copper-zinc alloy particles (mCu-Zn, 1500–2000 mesh) were purchased from Jiangsu Tianyi Superfine Metal Powder Co., Ltd (Huaian China). Sodium hydroxide (NaOH) and hydrogen peroxide (H_2_O_2_) were provided by Tianjin Hongda Chemical reagent. All materials were used as received without further purification.

### 2.2. Samples Preparation

#### 2.2.1. Particleboard Wood Flour (PWF) Processing and Chemical Modification

The particleboard was crashed into wood flour by high-speed pulverizer and sieved to 30 μm PWF (100 g). NaOH (30 g) and deionized water (1000 mL) were added to a dried flask and stirred (150–200 r/min) at 25 °C for 24 h to remove hemicellulose, pectin, and other impurities. Then, 30 wt.% H_2_O_2_ (17 mL) was added to the mixture and continuously stirred for 12 h (100–150 r/min) to remove the color composition of PWF. Finally, the product was filtered using a Buchner funnel and washed to neutral with deionized water. The obtained flour were dried at 80 °C for 12 h in a vacuum oven to remove the moisture.

#### 2.2.2. Preparation of micrometer copper-zinc alloy particles-reinforced wood plastic (mCu-Zn/PWF/PLA) composites

The composition of mCu-Zn/PWF/PLA composites is shown in Table 1. The mCu-Zn/PWF/PLA composites were fabricated by melt mixing with a twin-screw micro extruder. Before extrusion, PLA pellets and mCu-Zn were dried at 60 °C in an oven for at least 12 h to reduce the influence of moisture. The mCu-Zn, PWF, PLA pellets were mixed in a high-speed mixer for a uniform distribution, and subsequently fed into a co-rotating twin-screw extruder (SHJ-20, Nanjing GIANT Machinery Co., Ltd., Nanjing, China) for melt compounding. The rotation speeds of the screw and pelletizer were 25 rpm and 15 rpm, respectively, and the temperature profile of barrel sections from hopper to die was as follows: 145 °C, 155 °C, 165 °C, 175 °C, 175 °C, 170 °C. The extruded composites were cooled with water and pelletized, and then dried in an oven at 80 °C for 4 h to get rid of moisture.

#### 2.2.3. Preparation of Filaments and Printing Parameters for FDM 3D Printer

The masterbatch obtained by the above steps was added to the 3D printing consumables single screw extruder (MESI-25, Guangzhou Putong Experimental Analysis Instrument Co., Ltd., Guangzhou, China) to prepare a diameter of 1.75 ± 0.05 mm 3D printing filaments. The screw and the traction speeds were 15 rpm and 7 mm/min, respectively, the mould head temperature was 180 °C, and the cooling temperature was 40 °C. All kinds of filaments were successively printed by a FDM printer (FS-14, Guangzhou Flythinking Intelligent Technology Co., Ltd., Guangzhou, China) with 0.4 mm nozzle size. In this study, Cura software (Version 15.02.1, Ultimaker Corp., Utrecht, the Netherlands) was used to control the program parameters and command the printer. As shown in Figure 2, the printing path of the layer-by-layer sample was +45°/−45°, and the structural component of the sample was fabricated by the filament segments with a printing width of 0.4 mm, a layer height of 0.1 mm and shell thickness of 1.2 mm. In addition, the nozzle temperature of the printer was 200 °C, the platform temperature was 50 °C and the infill density was set at 100%. To demonstrate the printability of mCu-Zn/PWF/PLA filaments, some examples of printed parts with stable size and shapes are shown in Figure 1 and Figure 2, including a clearly patterned pen holder and a small dinosaur artwork that can be flexibly moved.

### 2.3. Characterization

Attenuated total internal reflectance-Fourier transform infrared spectroscopy (ATR-FTIR). The infrared spectra of the untreated PWF and alkali and peroxide treated PWF were recorded on attenuated total reflectance-Fourier transform infrared (ATR-FTIR) spectroscopy (Thermo Scientific Nicolet iS50, New York, NY, USA). The spectra were recorded from 400 to 4000 cm^−1^ with a resolution of 2 cm^−1^ and 32 scans.

#### 2.3.1. Mechanical Properties 

Experimental investigations were performed on standard test composites samples fabricated using the commercial FDM 3D printer (FS-14, Guangzhou Flythinking Intelligent Technology Co., Ltd., Guangzhou, China). Tensile tests and flexural tests of the composites samples were conducted according to ISO 527-3: 1995 and ISO 178: 2001, respectively, using an electronic universal testing machine (Shimadzu, AGS-X, Kyoto, Japan). For each group, more than five replicates were used in the test and the data were recorded as mean ± standard deviation. The tensile speed was 5 mm/min while the flexural test speed was 10 mm/min. Figure 2 shows the dimensions and printed path of the specimens for tensile and flexural tests.

#### 2.3.2. Dynamic Mechanical Thermal Analysis (DMA) 

The dynamic mechanical analyzer (NETZSCH, DMA 242C, Serb, Germany) was used to determine the storage modulus (*E*^’^), and loss factor (tan δ) of the composites as a function of temperature using a dual cantilever with specimen dimensions of 60 × 5 × 2 mm. The measurements were performed at a frequency of 1 Hz with a heating rate of 2 °C/min, and the temperature ranged from 35 to 100 °C.

#### 2.3.3. Thermal Gravity Analysis (TGA) 

The thermal stability and degradation of the composites were studied using a thermo gravimetric analyzer (NETZSCH, TG 209 F1, Serb, Germany). Samples of approximately 8 mg were heated from room temperature to 700 °C with 10 °C/min of heating rate and 20 mL/min of nitrogen gas flow.

#### 2.3.4. Differential Scanning Calorimetry (DSC) 

The thermal properties of composites were characterized using DSC (PE Co., Ltd., DSC 8000, Fremont, CA, USA). The sample (<8 mg) was heated from room temperature up to 200 °C at a rate of 10 °C /min, held for 5 min to remove the previous thermal history, cooled to 50 °C at a rate of 10 °C/min, and then reheated again to 200 °C with the same rate. Cold crystallization (*T_cc_*), corresponding melting (*T**_m_***), heat of fusion (Δ*H_m_*), degree of crystalline of PLA, and PWF/PLA and mCu-Zn/PWF/PLA composites were noted. The degree of crystallinity (*X**_c_*) was calculated according following equation:(1)$$X_{c}\;(\%)\;=\;\frac{\Delta H_{m}}{\Delta H_{m0}}\;\times 100\%$$
where Δ*H_m_* is the heat of fusion of the composites and Δ*H_m0_* is the melting enthalpy of pure PLA (93.7 J/g) [19].

#### 2.3.5. X-Ray Diffraction (XRD) 

X-ray diffraction measurement was carried out using an X-ray diffractometer (Bruker AXS, D8 ADVANCE, Karlsruhe, Germany) with CuKα radiation (λ = 1.542 Å) operated at 30 kV and 20 mA. Data were recorded in 2θ range of 10–80° at a scan rate of 5°/min.

#### 2.3.6. Scanning Electron Microscopy (SEM) and Energy Dispersive Spectrometer (EDS) 

The fracture surface structures of mCu-Zn/PWF/PLA composites were evaluated by SEM (XL-30E, Philip Electronic Optics Co., Ltd., Amsterdam, the Netherlands) with an acceleration voltage of 10 kV. All of the samples were fractured after immersion in liquid nitrogen. The brittle fracture surfaces were then gold-sprayed for the SEM observation. The elemental composition and content of the samples were measured by EDS (EDAX, Genesis, PA, USA).

#### 2.3.7. Surface Gloss Analysis 

The gloss reading was performed using a glossiness instrument (Shenzhen Sanenshi Technology Co., Ltd., YG-60, Shenzhen, China) with a 10 cm × 10 cm× 1 mm sample, with values expressed in Gloss Units (GU). This device measures the amount of light reflected from the surface of an object, which is then translated into a numerical scale. The measuring principle of this device is based on a light beam that strikes the object at a 60° [20,21,22,23]. The intensity of the reflected light is measured and compared to the reference value. Measurements were repeated five times for each specimen, and average values were calculated.

#### 2.3.8. Antibacterial Testing 

Plate count agar was used for evaluating the antibacterial properties of PLA, PWF/PLA, and mCu-Zn/PWF/PLA composite, in accordance with the ASTM E2149-10 test method [24]. The composites prepared from a 3D printer with a 10 × 10 mm film were sterilized for 30 min, at a temperature of 121 °C and a pressure of 103 KPa. The antibacterial effects of these mCu-Zn/PWF/PLA composites on a bacterial strain, *E. coli*, were investigated. Overnight cultures of *E. coli* derived from a single colony and cultivated in nutrient broth medium were used in the study. Each culture solution (1 mL) was inoculated into 9 mL of saline, resulting in a concentration of 3 × 10^5^–5 × 10^5^ colony forming units (CFUs/mL). These bacterial solutions with a concentration of 3 × 10^5^–5 × 10^5^ CFUs/mL were used for the antibacterial tests.

In a conical flask, 400 mg of shredded PWF/PLA and mCu-Zn/PWF/PLA composites were dispersed in 35 mL of saline, and then 5 mL of bacterial solutions with concentrations of 3 × 10^5^–5 × 10^5^ CFUs/mL were added. The bacterial solution containing pure PLA only served as controls. The conical flask was shaken in a constant temperature incubator shaker (Suzhou Peiying Experimental Equipment Co., Ltd., HZQ-F160, Suzhou, China) at 25 °C with a shaking speed of 300 rpm for 20 h. Aliquots (0.2 mL) of the obtained samples were placed on nutrient agar plates. Each sample was prepared separately in a triplicate. The inoculated plates were then held for 24 h at 37 °C ± 0.5 °C and the living cell bacteria colonies were carefully counted for evaluating the antibacterial activity. The inhibition rate was calculated according to the following equation [25]:(2)$$\mathrm{Inhibition}\;\mathrm{rate}\;(\%)\;=\;\frac{\mathrm{CFU}\;\mathrm{control}\;\mathrm{group}\;-\;\mathrm{CFU}\;\mathrm{experimental}\;\mathrm{group}}{\mathrm{CFUcontrol}\;\mathrm{group}}\;\times 100\%$$

## 3. Results and Discussion

### 3.1. FTIR Results

Figure 3 shows the Infrared spectra of untreated PWF and alkali and peroxide treated PWF. Comparing the two curves, the difference of absorption peaks was observed at 3348, 2905, 1739, 1239 cm^−1^ and 1058 cm^−1^. For the composites, the absorption peak at 3348 cm^−1^ is attributed to the stretching vibration of hydroxyl (–OH) in cellulose, hemi-cellulose, and lignin, and the absorption peak at 2905 cm^-1^ is attributed to saturated carbon (–C–H) [26]. The absorption peak at 1739 cm^−1^ is assigned to carbonyl from hemi-cellulose and lignin, the absorption peak at 1239 cm^−1^ is assigned to the –COO vibration of acetyl groups in hemicellulose, and the absorption peak at 1058 cm^−1^ is attributed to the C–O stretching of cellulose [26,27]. However, the absorption bands at 1739 cm^−1^ and 1239 cm^−1^ corresponding to the C=O and –COO functionalities of hemicellulose and lignin are absent in the alkali and peroxide treated PWF. The disappearance of these characteristic bands clearly indicated that the alkali and peroxide treatment significantly decreased the hemicellulose and lignin content, which would help to improve the thermal stability and interfacial compatibility of the composites [28].

### 3.2. Elemental Analysis 

The elemental analysis of the composition of the micrometer copper-zinc alloy particle was observed by EDS, and the results are shown in Figure 4. The micrometer particles were mainly composed of copper and zinc elements, of which the copper content was the highest, about 90.33%, followed by the zinc element, which was 8.46%, and finally the aluminum element, which was 0.81%. In addition, oxygen was also present at a content of 0.4%, which may be the result of chemical reaction of the metal exposed to air. It is well known that the copper-zinc alloy powder has a golden yellow color resembling gold, which can give wood plastic composite materials a unique color and high gloss. This is consistent with the results of the gloss test.

### 3.3. Mechanical Properties 

Figure 5 shows the effect of PWF content on the mechanical properties of PLA matrix. As seen in Figure 5a and Table 2, the tensile strength and elongation at break of PLA increased first and then decreased with increasing content of PWF. The pure PLA had a tensile strength of 43.79 ± 0.84 MPa and an elongation at break of 7.00%. The tensile strength of the PWF/PLA composites with PWF content of 5 wt.% reached the maximum tensile strength of 52.54 MPa (increment of 20%), while the elongation at break also reached a maximum of 8.52% (increment of 21.8%), which can be due to the fact that an appropriate amount of PWF restricted the mobility of the polymer chains and excess PWF led to the poor interfacial interaction between the PWF and the polymer matrix [29]. It is shown in Figure 5b that the flexural strength of every PWF/PLA composites in our experiments is higher than that of pure PLA, and the flexural modulus of PLA increases with increasing content of PWF. For PWF/PLA composites with a PWF content of 5 wt.%, the flexural strength reached a maximum of 80.66 MPa (increment of 23.7%), and the flexural modulus was 3.342 GPa (increment of 337%). These results demonstrated that the PWF improved the flexural strength at an appropriate concentration, due to its characteristic ratio of length to diameter. Moreover, the interaction between the filler and the PLA matrix was reduced with the addition of excess PWF, and this poor interaction was not effective in transferring the load to the filler [29,30]. Therefore, in the following studies, poly (lactic acid) was used as the matrix, and 5 wt.% particleboard wood flour and different contents of micrometer copper-zinc alloy particles were incorporated as reinforcement.

Figure 6 and Table 2 shows the effect of mCu-Zn content on the mechanical properties of mCu-Zn/PWF/PLA composites with 5 wt.% PWF. As seen in Figure 6a and Table 2, the tensile strength of mCu-Zn/PWF/PLA composites increased first and then decreased and elongation at break decreased with increasing content of mCu-Zn. The tensile strength reached a maximum of 56.64 MPa (increment of 7.6%), and the maximum elongation at break of 8.03% (decrease of 2.3%) was reached for the mCu-Zn/PWF/PLA composites with mCu-Zn content of 2 wt.%. In correspondence to reduced elongation for a ternary system composed of mCu-Zn, PWF, and PLA, the increasing content of mCu-Zn that improved the stiffness of the system weakened the interfacial regions between PWF/PLA and mCu-Zn, thus made the system more prone to crack propagation and illustrated more brittleness [31,32].

It is shown in Figure 6b that the flexural strengths of all mCu-Zn/PWF/PLA composite were higher than that of PWF/PLA composites, while a small amount of mCu-Zn would reduce the flexural modulus of the composites. The flexural modulus of the mCu-Zn/PWF/PLA composites increased first and then decreased with the mCu-Zn content continues to increase. For mCu-Zn/PWF/PLA composites with a mCu-Zn content of 2 wt.%, the flexural strength reached a maximum of 95.92 MPa (increment of 18.9%), and the flexural modulus was 2.24 GPa (decrease of 33.1%). This may be because mCu-Zn had a smaller size than PWF, and an appropriate amount of mCu-Zn can be uniformly filled into the PWF/PLA composites. Compared with other composites, mCu-Zn/PWF/PLA composites allowed effective stress transfer from wood plastic composites to mCu-Zn particles, and resulted in an improvement in the strength [32].

### 3.4. Dynamic Mechanical Properties

The thermomechanical properties of the composites were obtained by dynamic mechanical analysis (DMA). The storage modulus (*E*^’^), and the glass transition temperature (*Ttan*) of the samples are summarized in Table 3. Figure 7 shows the storage modulus and loss factor (tan δ) curves for the composites. Only one peak was observed in all the tan δ curves, demonstrating the homogeneity of the samples [33]. As shown in Figure 7a and Table 2, the storage modulus and loss factor of PWF/PLA composites increased first and then decreased. In detail, as the PWF content increased from 1 to 7, a decrease of *E*^’^ from 2248.84 to 1555.63 MPa at 35 °C, and of *Ttan* from 57.5 to 56.2 °C were observed for the resulting wood plastic composites. The reason may be that the storage modulus was found to decrease because of increased chain mobility of the wood cell wall polymeric components with the increase in temperature [34]. The viscoelastic behavior of wood was responsible for steep decrease of *E*^’^ values near the glass transition temperature of wood.

It was observed that higher storage modulus was shown by the samples treated with mCu-Zn/PWF/PLA in Figure 7b and Table 2. Compared with wood plastic composites, the storage modulus and glass transition temperature of the mCu-Zn/PWF/PLA were greatly improved. The storage modulus of PLA-1mCu-Zn was 2124.88 MPa and the glass transition temperature was 58.9 °C, which was 12.02% and 5.74% higher than that of PLA-5PWF, respectively. This might be because the wood plastic composites chains were inserted in between the metal particle and enhanced the interaction between them. A small amount of mCu-Zn in wood plastic composites could restrict the movement of the molecular chains, and therefore resulted in lower chain mobility, which was reflected by a higher *Ttan*. The energy was dissipated by friction between PWF, mCu-Zn particles and PLA matrix due to their interaction, which was the reason for the higher *E*^’^ [35,36].

### 3.5. Thermogravimetry Analysis

TGA profiles of the resulting composites are shown in Figure 8 and the TGA data including 5 wt.% loss temperature *T_on_*, the onset decomposition temperature [37]), the maximum rate of weight loss temperature (*T_max_*) and the end weight loss temperature (*T_end_*) are recorded in Table 3. In PWF/PLA composites, TGA showed two main degradation regions, where the weight loss of the composite material in the first region between 300 and 400 °C was mainly due to the degradation of the PLA matrix [38], and the higher temperature region comprised of the thermal degradation of hemicellulose, lignin, and cellulose in the PWF [39]. When the proportion of PWF increased, the *T_on_*, *T_max_,* and *T_end_* of the PWF/PLA composites gradually decreased, as shown in Figure 8a. The *T_on_*, *T_max_*, and *T_end_* of pure PLA were 25.2 °C, 21.4 °C, and 21.4 °C higher than those of the PLA-7PWF composites, respectively. The characteristic temperatures of PLA-1PWF, PLA-3PWF, and PLA-5PWF composites were intermediate to the pure PLA and PLA-7PWF. This phenomenon can be explained as follows: The thermal stabilities of PWF were lower than those of PLA, which led to the decrease of the thermal stability of the composites. Therefore, the increased proportion of the PWF led to an early decomposition of the composites [29]. 

It is shown in Figure 8b and Table 3 that the temperatures corresponding to the *T_on_*, *T_max_*, and *T_end_* of all the mCu-Zn/PWF/PLA composites were higher than those of the PLA-5PWF. A larger difference of 13.4 °C, 12.3 °C, and 13.0 °C was found between PLA-1mCu-Zn and PLA-5PWF composites. This implies that the addition of mCu-Zn can delay the initial stage of thermal decomposition of wood plastic composites. Moreover, it was noted that addition of mCu-Zn slightly increased the residue at 500 °C of wood plastic composites. This was mainly due to the interaction of the mCu-Zn/PWF/PLA composite components, the good thermal conductivity of mCu-Zn, and also the fact that good dispersion state of mCu-Zn clusters can ensure thermal conductive efficiency. Therefore, the thermal stability of wood plastic composites can be improved by incorporating mCu-Zn into composites [40].

### 3.6. Differential Scanning Calorimetry Analysis

The influence of PWF and mCu-Zn on crystallization behavior of PLA was investigated by DSC. Figure 9 shows the DSC thermograms of PLA, PWF/PLA, and mCu-Zn/PWF/PLA composites in the second heat scan after fast cooling at 10 °C/min from the molten state, and the relevant data are listed in Table 4. It can be seen from Figure 9a and Table 4 that the pure PLA showed three transitions during the heating process, including (1) heat jump corresponding to a glass transition at 62.71 °C, (2) cold-crystallization exothermal peak occurred at 105.41 °C, and (3) melting endothermal peak at 167.92 °C. 

Moreover, the enthalpy of fusion (Δ*H_m_*) of all the wood plastic composites was higher than that of the pure PLA, which implied that the crystallinity of PLA was increased by addition of PWF in it. This increase of crystallinity reinforced the presumption that lignocellulosic fibers cause nucleation [41], and thus the modulus of PWF/PLA composites would be increased. That is consistent with the results obtained by the mechanical properties. In addition, the crystallinity of PWF/PLA wood plastic composites increased first, then decreased, and then increased with the increase of PWF. Thus, there were trends in our data to suggest that when 1 wt.% PWF was added, PWF acted as a nucleating agent to promote the dominant position of PLA crystals, while when 3–5 wt.% PWF was added, PWF hindered the movement of PLA segments and reduced the proportion of PLA crystals. In the dominant position, with the final addition of 7 wt.% PWF, the former acted more than the latter, resulting in a higher crystallinity of the PLA [42]. 

It is shown in Figure 9b and Table 4 that the addition of mCu-Zn in PWF/PLA composites did not result in a noticeable change for the glass transition of wood plastic composites. The cold crystallization temperature (*T_cc_*) of wood plastic composites was improved by the incorporation of mCu-Zn over 1 wt.%, and the crystallinities of all the mCu-Zn/PWF/PLA composites were similar to that of wood plastic composites. Two main factors control the crystallization of polymeric composites systems: (i) The additives hinder the migration and diffusion of polymer molecular chains to the surface of the growing polymer crystal in the composites, thus providing a negative effect on polymer crystallization which results in a decrease in the *T_cc_*; (ii) the additives have a nucleating effect which gives a positive effect on polymer crystallization and leads to an increase in *T_cc_* [42]. However, in this case, the *T_cc_* of mCu-Zn/PWF/PLA composites has improved. It may be concluded that in the presence of mCu-Zn, the crystallinity of the composites increased, which helped to improve the mechanical properties of the composites.

### 3.7. X-Ray Diffraction Spectrum Analysis

The XRD spectra of mCu-Zn and mCu-Zn/PWF/PLA composites are shown in Figure 10. In Figure 10a, the XRD spectrum of mCu-Zn showed three peaks at 31.9°, 34.6°, and 36.4° corresponding to the planes of zinc, and peaks at 43.9°, 50.6°, 75.1° corresponding to (111), (200), and (220) planes of copper, which are in agreement with the JCPDS, copper file No. 04-0836 [12]. Moreover, no other crystalline peak was detected, which implies the high purity of the mCu-Zn. In addition, the peaks corresponding to the mCu-Zn appear in the XRD spectra of all mCu-Zn/PWF/PLA composites. As can be seen in Figure 10b, the diffraction peaks at 2θ = 16.8° and 19.6°, are corresponding to the characteristic peaks of the (110) and (020) planes of PLA, respectively [43]. The result shows the positions of diffraction peaks of PLA in all composites substantially unchanged after the PWF and mCu-Zn being added to polymer matrix, which indicates that the crystal type of PLA in mCu-Zn/PWF/PLA composites did not change.

### 3.8. Morphology of the Composites

The morphology of the fracture surface of the composites was investigated through SEM. The SEM micrographs on the fracture surfaces of the pure PLA and PWF/PLA composites are shown in Figure 11a,b. In Figure 11a, it shows that the fracture surface of the pure PLA was relatively flat and no voids were observed. Therefore, the PLA materials showed brittleness. Additionally, Figure 11b illustrates that 5 wt.% PWF was well bonded to PLA matrix and only a few gaps in the fiber/polymer interface was found, indicating good adhesion between the PWF and the PLA matrix. Moreover, PWF was basically encompassed by the PLA matrix, which further proved that the bonding of the PWF and PLA matrix was in a good condition after melt extrusion. The uniform stress distribution of the continuous PLA matrix to the dispersed PWF phase provided excellent mechanical properties [31,44].

SEM photographs of fractured surface of the mCu-Zn/PWF/PLA composites are shown in Figure 11c–f. Adding mCu-Zn into the wood plastic composites revealed that there was substantially no gap between the PWF and the PLA in the fracture surface of the composites. However, full PWF shedding was observed in PLA-1mCu-Zn and PLA-4mCu-Zn, indicating that a small amount or excessive mCu-Zn had little interaction with increasing PWF and PLA. This means that the mechanical properties of PLA-1mCu-Zn and PLA-4mCu-Zn were not good compared to other tested composites. Simultaneously, many mCu-Zn particles existed on the surface of PLA-4mCu-Zn in Figure 11f, which indicates that excessive mCu-Zn led to poor compatibility of the reinforcement and the matrix and lowered the properties of the composites. In addition, PLA-2mCu-Zn and PLA-3mCu-Zn composites showed smoother fracture surfaces, with less pulled-out wood flour from the fractured surface in Figure 11d,e, in comparison with the other tested composites. This might be due to the fact that the combination of the mCu-Zn and PWF increased the interaction with PLA matrix [31].

Moreover, it can be seen that there was not any obvious agglomeration of mCu-Zn on the fracture surface besides PLA-4mCu-Zn, which indicates a uniform distribution of mCu-Zn within the PWF/PLA composites, as confirmed by the gloss testing results. The uniform distribution of the mCu-Zn is critical to ensure good mechanical properties of the mCu-Zn/PWF/PLA composites.

### 3.9. Surface Gloss Analysis 

Surface gloss of the mCu-Zn/PWF/PLA composites are shown in Figure 12. The mCu-Zn/PWF/PLA composites gloss ranged from 38 to 73 GU (gloss units). The gloss of mCu-Zn /PWF/PLA composites increased first and then decreased with the increase of mCu-Zn. The gloss of PLA-5PWF was 5 GU. After adding 1 wt.% mCu-Zn and 2 wt.% mCu-Zn, the gloss of the composites reached 59.57 GU and 62.13 GU, respectively, which increased by 1091.4% and 1142.6%. Because mCu-Zn had excellent reflective ability, a small amount of mCu-Zn can be evenly distributed on the surface of the composites, resulting in a greater increase in the gloss of the material. However, the gloss of PLA-3mCu-Zn and PLA-4mCu-Zn began to decrease, which was 54.65 and 38.53 GU, respectively. This illustrates that excess mCu-Zn caused the surface of the composites to be unsmooth, resulting in a decrease in gloss.

### 3.10. Antibacterial Activity 

The antibacterial activity against *E. coli* was studied in mCu-Zn/PWF/PLA composites as shown in Figure 13. The bacteria were incubated in a growth medium with the pure PLA and the composites, and the antibacterial properties of samples were tested using the antibacterial testing standard (ASTM E2149-10) [24]. The number density of bacterial colony for pure PLA was high, with no conceivable antibacterial activity. The inhibition rates of the PLA-5PWF was 27.66% as shown in Figure 13, which showed that wood plastic composites did not have excellent antibacterial effects. However, the inhibition rate of the PLA-2mCu-Zn composites was 90.43%, which showed that the mCu-Zn/PWF/PLA composites had excellent antibacterial effects on *E. coli*. It could be due to the replacement of essential ions and the blocking of protein functional groups by copper and zinc, enzyme inactivation and production of free radicals by membrane bound copper and zinc, and the destruction of normal membrane integrity [12]. These results indicated that the antibacterial effect of the composites came from the strong activity of the mCu-Zn component. 

### 3.11. Application in 3D Printing

Figure 14 shows images of the 3D printing samples made from the composite materials. In the FDM 3D printing processes, the printer melts the thermoplastic material through a heater, then the nozzles extrude the filaments and stack them layer-by-layer, eventually turning the blueprint in the computer into a physical object [14]. Compared with traditional processing methods, FDM 3D printing greatly shortens the product development cycle, exhibits high material utilization rate, increases productivity and lowers production costs [45]. 

Wood flour/poly (lactic acid) composites have becen widely applied in the field of 3D printing due to their excellent mechanical properties and complete degradability. However, the poor thermal stability and rough appearance of these composites limit their wide application in a variety of fields. The purpose of this study is to investigate the possibility of mCu-Zn/PWF/PLA as an antibacterial composite for 3D printing and compare its performances with those of PWF/PLA composites. As shown in Figure 14a–c, the wood plastic composites for 3D printing has a natural wood color, but the printed dumbbell-shaped tensile samples have noticeable texture, and the printed chair looks dull and rough. This is because the presence of so many fibers in the wood plastic composites results in unevenness in printing as well as a rough surface of the product. Post processing is often required to improve the appearance of the final product. Compared with wood plastic composites filaments for 3D printing, mCu-Zn/PWF/PLA composites have a unique metallic luster, a smooth surface, and a well texture, as shown in Figure 14d–f. The increase in gloss is primarily attributed to the fact that mCu-Zn is uniformly filled inside and on the surface of the wood plastic composites, which makes the surface of the printed product smoother. Moreover, mCu-Zn/PWF/PLA composites have high mechanical properties and thermal stability, as well as excellent bacteriostatic effect. 

The composites prepared in this study demonstrate many advantages, including environmental friendliness (no requirement for post-treatment such as spray painting), low cost, excellent antibacterial properties, and high gloss, which make them promising materials for FDM 3D printing technology.

## 4. Conclusions

The mCu-Zn/PWF/PLA wood plastic composite filaments with high gloss and antibacterial properties for FDM were prepared by melt blending process. First, the waste wood chips of the particleboard (PWF) were treated with alkali and hydrogen peroxide. The FTIR showed that the content of hemicellulose and lignin in the PWF was significantly reduced. Second, the effect of different contents of PWF on the properties of the PLA material was investigated. The results show that when the PWF content was 5 wt.%, the mechanical properties of the composites reached a maximum. Subsequently, the effect of different contents of mCu-Zn on the properties of PWF/PLA wood plastic composites with 5 wt.% PWF was studied. The addition of 2 wt.% mCu-Zn reinforcement can significantly improve the gloss and antibacterial effect of PWF/PLA wood plastic composites. Moreover, TGA and DSC results show that the addition of mCu-Zn can improve the thermal properties of wood plastic composites and promote the crystallization of poly (lactic acid). Finally, this novel high gloss and antibacterial wood plastic composites successfully produced personalized dinosaur artwork, exquisite pen holders, and furniture models through FDM 3D printing.

## Figures and Tables

**Figure 1 polymers-12-00621-f001:**
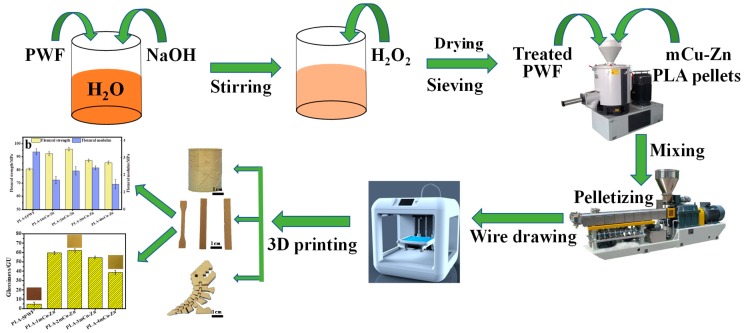
Flowchart showing the chemical modification of particleboard wood flour (PWF), preparation and characterization of micrometer copper-zinc alloy particles-reinforced particleboard wood flour/poly (lactic acid)(mCu-Zn/PWF/PLA) composites for 3D printing.

**Figure 2 polymers-12-00621-f002:**
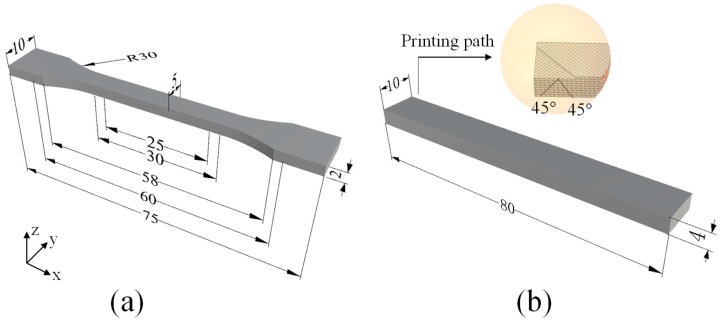
Standard specimens for mechanical testings (mm): (**a**) Dog bone-shaped specimen for the tensile test, (**b**) rectangular specimen for the flexural test.

**Figure 3 polymers-12-00621-f003:**
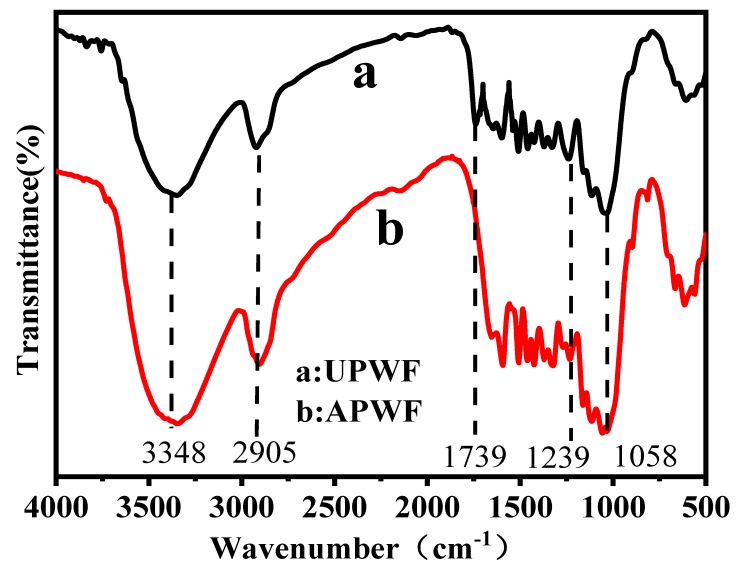
Infrared spectra of (**a**) untreated PWF, (**b**) alkali and peroxide treated PWF.

**Figure 4 polymers-12-00621-f004:**
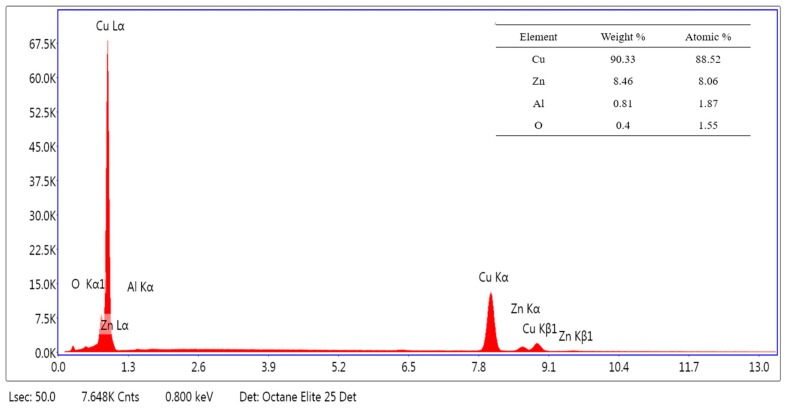
Elemental analysis of micrometer copper-zinc alloy particles.

**Figure 5 polymers-12-00621-f005:**
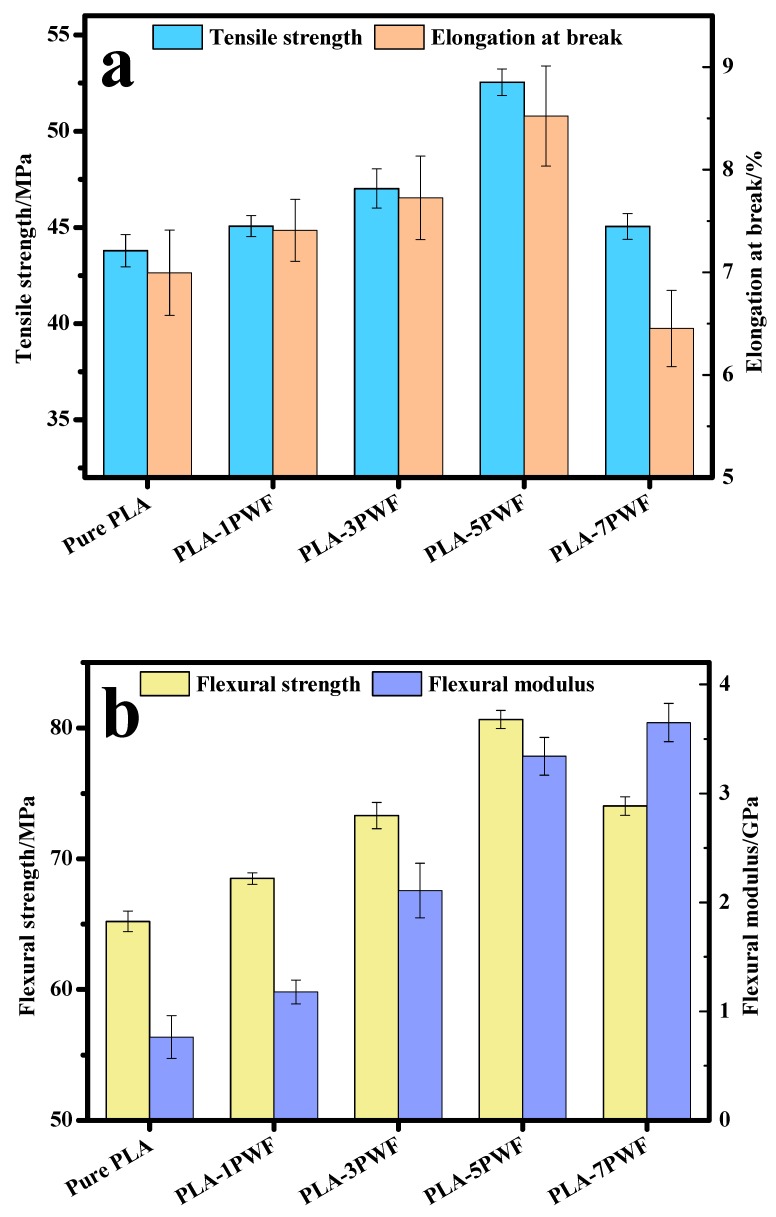
Tensile properties (**a**) and flexural properties (**b**) of PWF/PLA composites samples.

**Figure 6 polymers-12-00621-f006:**
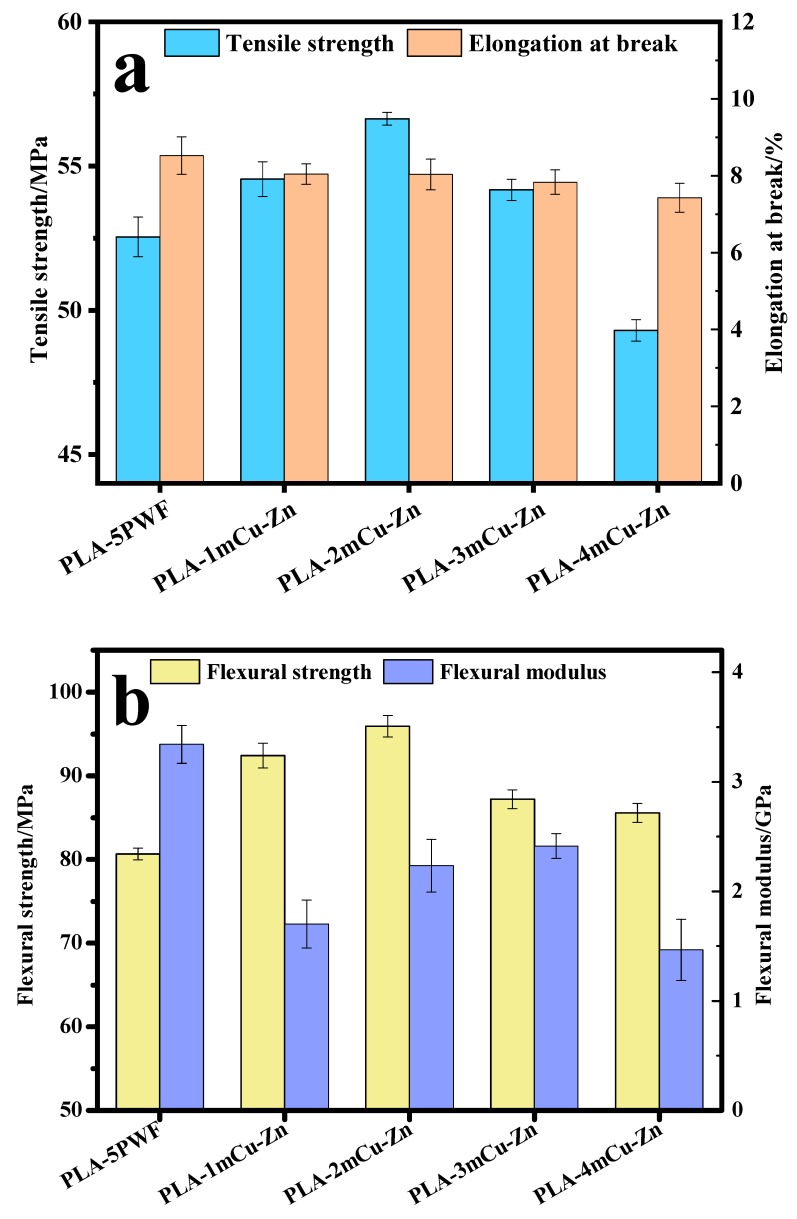
Tensile properties (**a**) and flexural properties (**b**) of mCu-Zn/PWF/PLA composites samples with 5 wt.% PWF.

**Figure 7 polymers-12-00621-f007:**
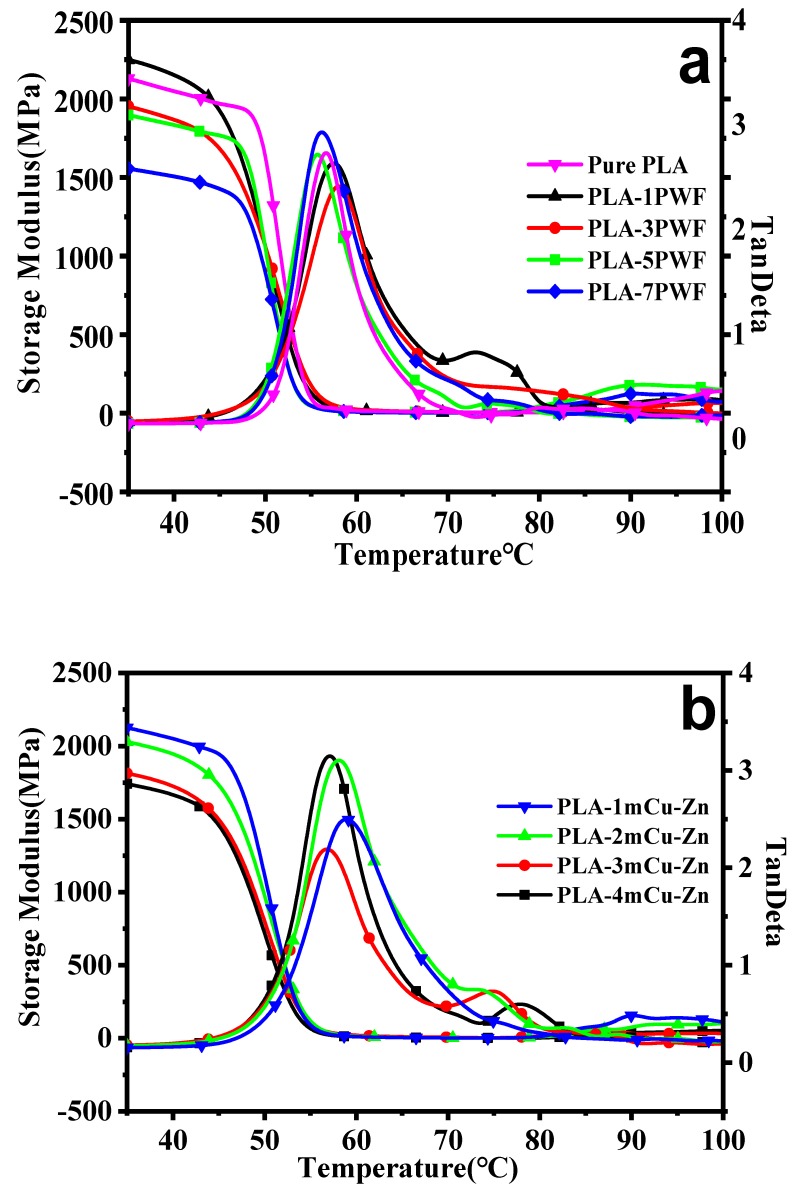
Dynamic mechanical thermal analysis (DMA) curves of PWF/PLA composites (**a**) and mCu-Zn/PWF/PLA composites with 5 wt.% PWF (**b**).

**Figure 8 polymers-12-00621-f008:**
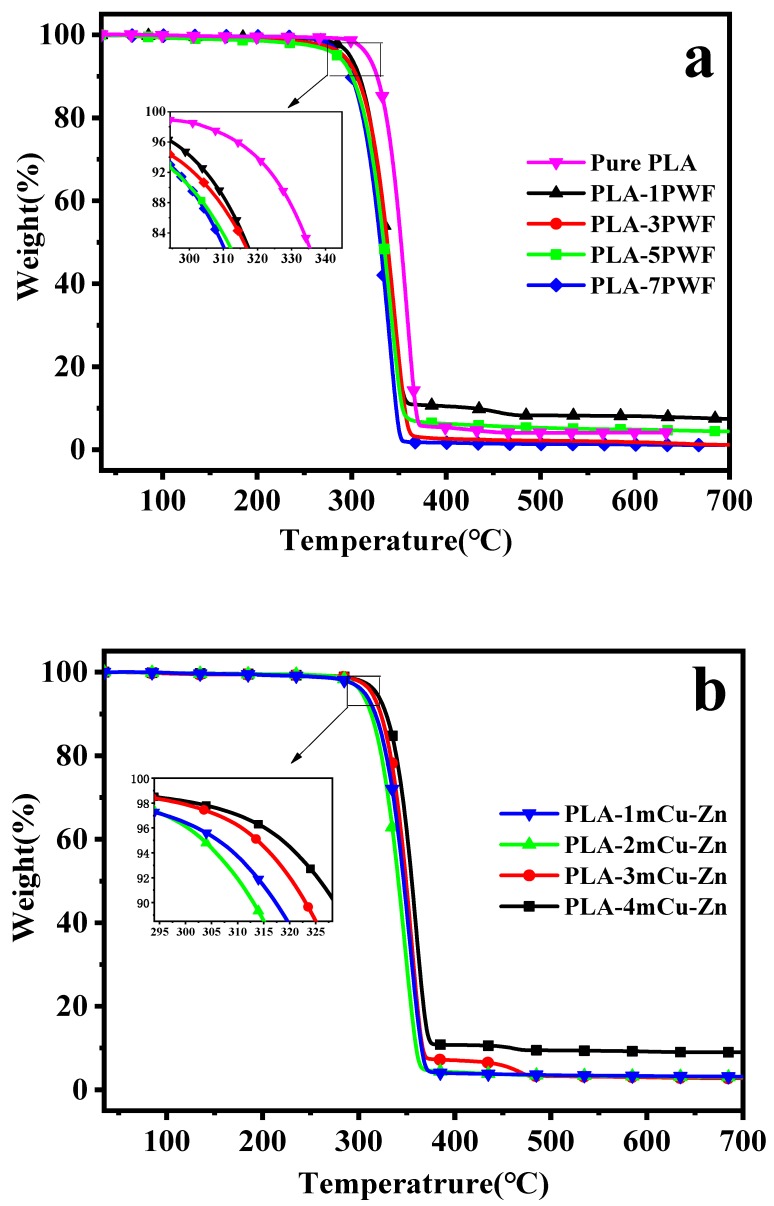
Thermal gravity analysis (TGA) curves of PWF/PLA composites (**a**) and mCu-Zn/PWF/PLA composites with 5 wt.% PWF (**b**).

**Figure 9 polymers-12-00621-f009:**
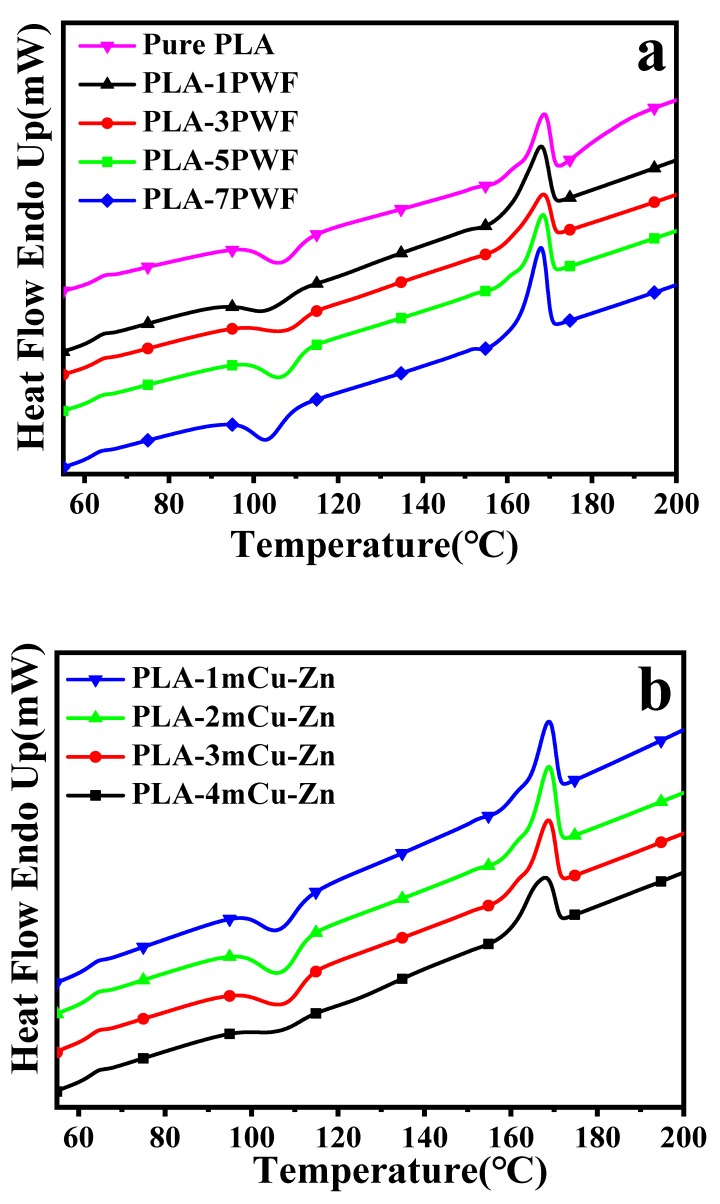
Differential scanning calorimetry (DSC) curves of PWF/PLA composites (**a**) and mCu-Zn/PWF/PLA composites with 5 wt.% PWF (**b**).

**Figure 10 polymers-12-00621-f010:**
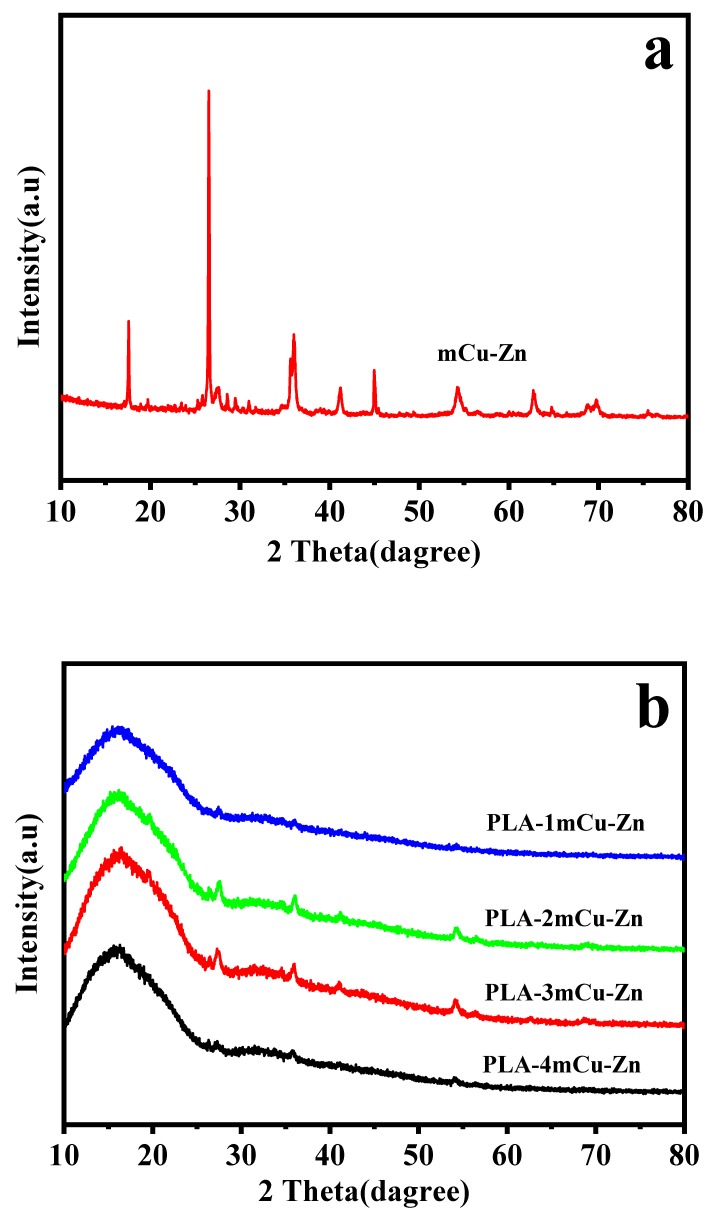
The X-Ray diffraction (XRD) spectra of mCu-Zn (**a**) and mCu-Zn/PWF/PLA composites with 5 wt.% PWF (**b**).

**Figure 11 polymers-12-00621-f011:**
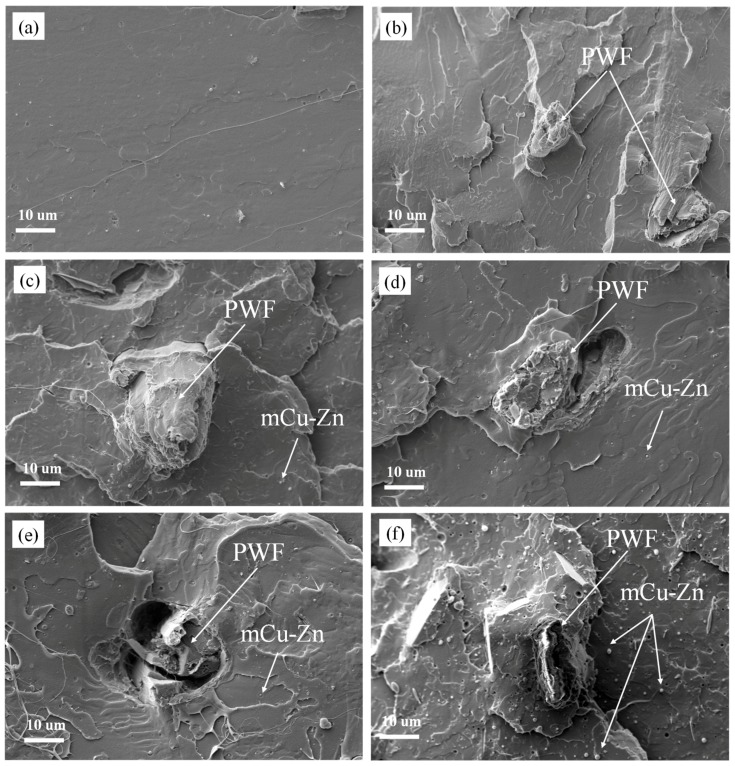
Scanning electron microscopy (SEM) images of the fracture surface of the composites: (**a**) Pure PLA, (**b**) PLA-5PWF, (**c**) PLA-1mCu-Zn, (**d**) PLA-2mCu-Zn, (**e**) PLA-3mCu-Zn, (**f**) PLA-4mCu-Zn.

**Figure 12 polymers-12-00621-f012:**
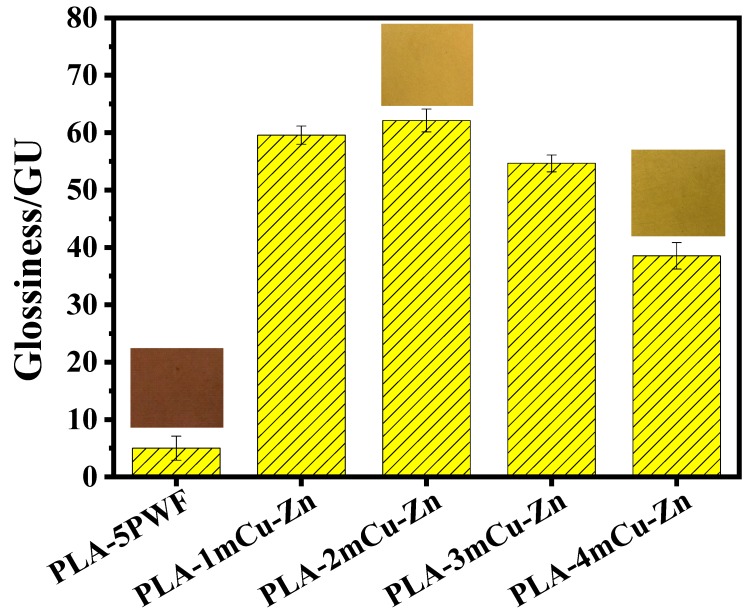
The gloss value of PLA-5PWF and mCu-Zn/PWF/PLA composites with 5 wt.% PWF.

**Figure 13 polymers-12-00621-f013:**
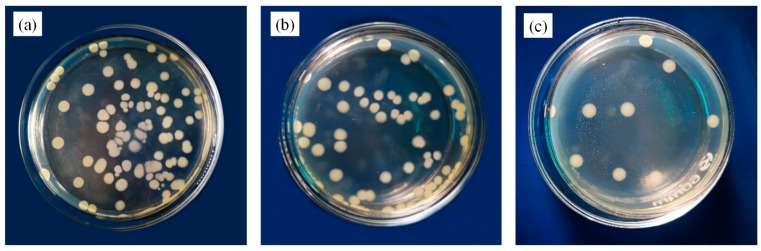
Images showing mCu-Zn/PWF/PLA composites against *E. coli*: (**a**) Pure PLA; (**b**) PLA-5PWF; (**c**) PLA-2mCu-Zn.

**Figure 14 polymers-12-00621-f014:**
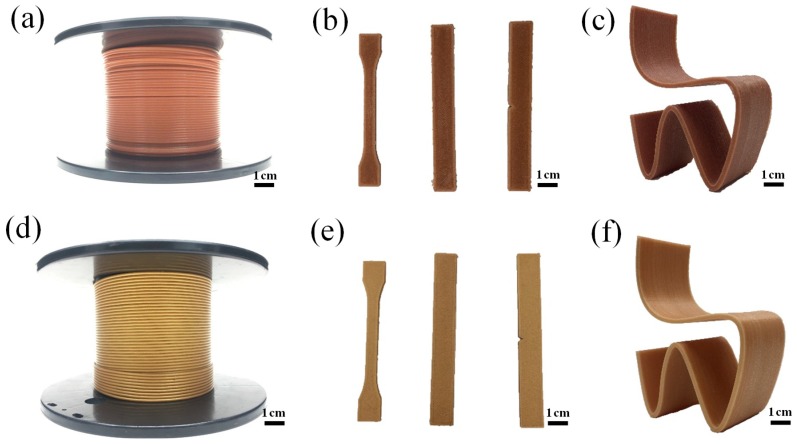
Filaments, mechanical testing samples and chair models made by fused deposition modeling (FDM) 3D printing using the composites: (**a**,**b**,**c**) PLA-5PWF; (**d**,**e**,**f**) PLA-2mCu-Zn.

**Table 1 polymers-12-00621-t001:** Compositions of particleboard wood flour/poly (lactic acid) (PWF/PLA) and mCu-Zn/PWF/PLA composites.

Sample	PWF (%)	**mCu-Zn** (%)	PLA (%)
PLA-1PWF	1	—	99
PLA-3PWF	3	—	97
PLA-5PWF	5	—	95
PLA-7PWF	7	—	93
PLA-1mCu-Zn	4.95	1	94.05
PLA-2mCu-Zn	4.9	2	93.1
PLA-3mCu-Zn	4.85	3	92.15
PLA-4mCu-Zn	4.8	4	91.2

**Table 2 polymers-12-00621-t002:** Mechanical properties of PWF/PLA and mCu-Zn/PWF/PLA composites.

Sample	Tensile Strength (MPa)	Elongation at Break (%)	Flexural Strength (MPa)	Young’s Modulus (GPa)
Pure PLA	43.79 ± 0.84	7.00 ± 0.41	65.21 ± 0.84	0.76 ± 0.20
PLA-1PWF	45.07 ± 0.55	7.41 ± 0.30	68.50 ± 0.79	1.18 ± 0.11
PLA-3PWF	47.02 ± 1.02	7.73 ± 0.40	73.31 ± 0.44	2.11 ± 0.25
PLA-5PWF	52.54 ± 0.69	8.52 ± 0.49	80.66 ± 1.00	3.34 ± 0.17
PLA-7PWF	45.05 ± 0.67	6.45 ± 0.37	74.04 ± 0.70	3.65 ± 0.18
PLA-1mCu-Zn	54.55 ± 0.60	8.04 ± 0.26	92.42 ± 1.17	1.70 ± 0.22
PLA-2mCu-Zn	56.64 ± 0.22	8.03 ± 0.40	95.92 ± 1.28	2.24 ± 0.24
PLA-3mCu-Zn	54.17 ± 0.37	7.83 ± 0.32	87.22 ± 1.13	2.41 ± 0.11
PLA-4mCu-Zn	49.30 ± 0.38	7.43 ± 0.38	85.57 ± 1.13	1.47 ± 0.28

**Table 3 polymers-12-00621-t003:** DMA and TGA data of PWF/PLA and mCu-Zn/PWF/PLA composites.

Sample	DMA			TGA		
	*Ttan* (°C)	*E*^’^ at 35 °C (MPa)		*T_on_*(°C)	*T_max_* (°C)	*T_end_*(°C)
Pure PLA	56.6	2128.35		334.9	358.6	370.1
PLA-1PWF	57.5	2248.84		316.6	345.8	358.3
PLA-3PWF	58.0	1951.70		314.3	343.7	355.3
PLA-5PWF	55.7	1896.91		312.0	341.8	354.1
PLA-7PWF	56.2	1555.63		309.7	337.2	348.7
PLA-1mCu-Zn	58.9	2124.88		325.4	354.1	367.1
PLA-2mCu-Zn	58.1	2027.33		319.5	348.9	361.7
PLA-3mCu-Zn	56.8	1813.43		330.0	356.2	367.6
PLA-4mCu-Zn	57.1	1741.23		335.6	360.7	371.4

**Table 4 polymers-12-00621-t004:** DSC data of PWF/PLA and mCu-Zn/PWF/PLA composites.

Sample	Glass Transition Temperature *T_g_* (°C)	Crystallization Temperature *T_cc_* (°C)	Melting Temperature *T_m_* (°C)	Melting Enthalpy Δ*H_m_* (J/g)	Crystallinity *X_C_* (%)
Pure-PLA	62.71	105.41	167.92	30.11	32.1
PLA-1PWF	62.45	102.15	167.32	36.31	38.8
PLA-3PWF	62.51	106.92	167.83	31.78	33.9
PLA-5PWF	62.40	105.72	167.82	33.30	36.1
PLA-7PWF	62.31	102.32	167.36	37.51	40.0
PLA-1mCu-Zn	62.45	105.80	167.95	35.77	38.2
PLA-2mCu-Zn	62.51	105.93	168.17	31.81	33.9
PLA-3mCu-Zn	62.57	106.80	168.10	33.41	35.7
PLA-4mCu-Zn	62.67	106.08	167.09	32.71	34.9

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
