# Peer review of "Micrometer Copper-Zinc Alloy Particles-Reinforced Wood Plastic Composites with High Gloss and Antibacterial Properties for 3D Printing"

_polymers, 2020, doi:10.3390/polym12030621_

Round 1
Reviewer 1 Report
The paper is well written and easy to follow. Analyses are well described and the authors have performed a relevantly high number of measurements. Therefore I do believe this paper deserves publication in Polymers.
I ask the authors to complete the literature review in the Introduction section with the recent review article by Mazzanti et al. that deals with the same subject:
Mazzanti V., Malagutti L., Mollica F. (2019). FDM 3D printing of polymers containing natural fillers: A review of their mechanical properties. POLYMERS, vol. 11, p. Article number 1094, ISSN: 2073-4360, doi:10.3390/polym11071094
Moreover, in Fig. 5a, the "I"'s should be removed in the x axis labels from "PLA3IPWF" and "PLA5IPWF". Also, in Figs. 5b and 6b the flexural modulus should be measured in GPa, not MPa. Finally, Figs. 7a and 7b should appear close together.
Reviewer 2 Report
The aim of the research was to prepare poly(lactic acid) composites reinforced with particleboard wood flour and micrometre copper-zinc alloy particles in the melt blending process. The paper is an interesting study that can contribute to the scope of the Polymers. However, it contains a lot of mistakes that need to be corrected.
- Title is not linguistically correct. It is not known what is reinforced, whether particles or composites. Please correct.
- “In this work, high gloss and antibacterial micron copper-zinc alloy particles reinforced particleboard wood flour/polylactic acid (mCu-Zn/PWF/PLA) wood plastic composites for 3D printing were prepared by a melt blending process”
Please reduce the use of abbreviations in the abstract to a minimum. Please rearrange this sentence to make it understandable correctly.
- “E. coli” – please use the entire name at least in the first place of use. Same for “S. aureus”. Please correct.
- “Polylactic acid is a biodegradable plastic with similar properties to many petroleum-based plastics”.
Generally biodegradability is a polymer property, not a plastic. In this context the word polymer will be more appropriate. Please correct.
- “Micron” – according to the SI system, micrometre (international spelling) or micrometer (American spelling) is used. Please correct throughout the text.
- “polylactic acid”, should be “poly(lactic acid)”.
According IUPAC terminology when the name of the monomer comprises two or more words, parentheses should be used. Please correct throughout the text.
- Unit symbol should be written according to the principles of SI using subscripts and italics, for example Tcc. Please correct throughout the text.
- “wood plastic composites” or “wood-plastic composites”. The text should be unified. Please correct throughout the text.
Line 41: “polylactic” – acid is missing. Please correct.
Line 71: “used 3D Printing technologies” – typo mistake.
“Printing technology” is not a proper name. Please correct.
Line 90: “g/10 min-@210 °C/2.16 kg”, Please correct typo mistake.
Line90/91: “( PWF)” – typo mistake.
Please remove a space after parentheses.
Line 130: It should probably be Fig. 2.
Line 242: “polymer blend” – the authors probably meant “polymer matrix”. Please correct.
Line 250-252: The matrix is a polymer. Particleboard wood flour in an amount of 5% is also a composite reinforcement. Please correct.
Line 268, 278, 366 and 425: “between wood plastic matrices and mCu-Zn particles”.
Wood and matrix are composite, so you can use a slash as a matrix connector with wooden reinforcement. (“between wood/plastic (or more precisely PWF/PLA) and mCu-Zn particles”). Please correct.
Line 335/336: “This was mainly attributed to mCu-Zn/PWF/PLA composites matrix interaction”.
Did the authors mean: “This was mainly due to the interaction of the mCu-Zn/PWF/PLA composite components”. Please correct.
Line 278, also 185, 242, 325, 326, 328: “Compared with other blends, mCu-Zn/PWF/PLA 278 composites”.
In polymer chemistry, we divide mixtures into polymer blends (two different polymers) and polymer composites. In the above case, if I understand correctly, we have a polymer composite and I would ask the authors to maintain this terminology. Please correct.
Line 289: The authors observe the glass transition temperature using DMA. I would suggest to use the appropriate symbols (Ttan by DMA, Tg by DSC), because the values of glass transition temperature observed using DMA are usually higher than those observed using DSC. Please correct throughout the text.
Line 319-322: “In PWF/PLA composites, TGA showed two main degradation regions, where the first region comprised of the thermal degradation of hemicellulose, lignin and cellulose in the PWF [36], and the higher temperature region corresponded to the depolymerization of the PLA [37].”
The first stage of mass loss refers to PLA (Tmax about 350), the second to wood components, which are more stable (the first-derivative TG (DTG) is more clear in this case). Please correct.
Line 353: Rather, DSC curves show that the glass transition temperature is between 60 and 65 °C. Could the authors insert Tg values into the Table 4?
Figure 11: “SEM images of the micro morphology of composites”.
Images can be on a micro-scale, but not morphology. Please correct.
Round 2
Reviewer 2 Report
Thank you for considering my comments.
Line 98: “@”; What does this sign mean? It's probably a mistake when converting text.
Line 360: “heat jump corresponding to a glass transition at 59.97 °C”.
The table has a different value for Tg. Please correct.
Table 4: Degrees should be in brackets like other units.
Author Response
Dear Reviewers:
Thank you very much for your kind attention and careful review for our manuscript entitled ‘Micrometer Copper-Zinc Alloy Particles-Reinforced Wood Plastic Composites with High Gloss and Antibacterial Properties for 3D Printing’. The following is the feedback.
- [Line 98: “@”; What does this sign mean? It's probably a mistake when converting text.]
Response: “7 (g/10 min–@210 °C/2.16 kg)” have been modified to “7 g/10 min (at 210 °C, load 2.16 kg)”, For details, please see line 98 in the revised manuscript.
- [Line 360: “heat jump corresponding to a glass transition at 59.97 °C”. The table has a different value for Tg. Please correct.]
Response: In line 358 of the revised manuscript, “heat jump corresponding to a glass transition at 59.97 °C” have been modified to “heat jump corresponding to a glass transition at 62.71 °C”.
- [Table 4: Degrees should be in brackets like other units.]
Response: In Table 4 of the revised manuscript, “Tg/°C”, “Tcc/°C”, “Tm/°C” and “ΔHm/(J/g)” have been modified to “Tg (°C)”, “Tcc (°C)”, “Tm (°C)” and “ΔHm (J/g)”. For details, please see line 388 in the revised manuscript.
In general, all of the corrections we made are shown with a red font form in a copy of the new revised manuscript.
